# Congenital Zika Syndrome: Insights from Integrated Proteomic and Metabolomic Analysis

**DOI:** 10.3390/biom15010032

**Published:** 2024-12-30

**Authors:** Leticia Gomes-de-Pontes, Lucila Akune Barreiros, Lillian Nunes Gomes, Ranieri Coelho Salgado, Sarah Maria da Silva Napoleão, Paulo V. Soeiro-Pereira, Saulo Duarte Passos, Antonio Condino-Neto

**Affiliations:** 1Department of Immunology (LIH), Institute of Biomedical Sciences, University of São Paulo, Av. Prof. Lineu Prestes, 2415-Butantã, São Paulo 05508-000, SP, Brazil; lucila.barreiros@alumni.usp.br (L.A.B.); lillian.gomes@usp.br (L.N.G.); ranierics@gmail.com (R.C.S.); smnapoleao@gmail.com (S.M.d.S.N.); 2Department of Pathology, Federal University of Maranhão, São Luís 65065-545, MA, Brazil; paulo.soeiro@ufma.br; 3Infectious Pediatric Laboratory, Medicine School of Jundiaí, Jundiaí 13202-550, SP, Brazil; sauloduarte@uol.com.br

**Keywords:** Zika infection, extracellular, vesicles, proteome, metabolome

## Abstract

**Background:** In this study, we investigated the role of extracellular vesicles (EVs) in the pathogenesis of Congenital Zika Syndrome (CZS). Previous studies have highlighted the role of EVs in intercellular communication and the modulation of biological processes during viral infections, motivating our in-depth analysis. Our objective was to identify specific molecular signatures in the EVs of patients with CZS, focusing on their potential as biomarkers and on cellular pathways affected by the infection. **Methods:** We conducted advanced proteomic and metabolomic analyses using mass spectrometry for protein and metabolite identification. EVs were isolated from CZS patient samples and control groups using Izon qEV size-exclusion chromatography columns. **Results:** The analyzed EVs presented distinct molecular profiles in patients with CZS. Proteomic analysis revealed significant alterations in specific proteins, suggesting involvement in the PI3K-AKT-mTOR pathway, while metabolomics highlighted metabolites related to critical processes in Zika virus pathogenesis. These findings suggest a key role for the PI3K-AKT-mTOR pathway in regulating cellular processes during infection and indicate the involvement of EVs in intercellular communication. Additionally, the results identified potential biomarkers capable of aiding early diagnosis and assessing disease progression. **Conclusions:** This study demonstrates that EVs play a crucial role in intercellular communication during Zika virus infection. The identification of specific alterations in the PI3K-AKT-mTOR pathway highlights a possible therapeutic target, providing new opportunities for the development of more effective treatment strategies for CZS. Our findings significantly advance the understanding of CZS and underscore the need for further investigations using advanced techniques to validate and explore these potential molecular targets.

## 1. Introduction

Zika virus infection during pregnancy poses a significant risk to fetal development, leading to severe birth defects such as microcephaly and other neurological abnormalities [1]. In addition, adults infected with Zika can experience a range of neurological complications, including encephalitis, meningitis, and Guillain–Barré syndrome. Despite ongoing research, the precise mechanisms underlying these severe manifestations remain elusive. Currently, there are no specific vaccines or treatments available for Zika virus infection [2,3].

The increasing global spread of the virus, driven by factors like viral virulence, vulnerable populations, and efficient vector transmission, underscores the urgent need for effective prevention and treatment strategies. The primary vectors for Zika and dengue viruses are *Aedes aegypti* and *Aedes albopictus* mosquitoes. Female mosquitoes become infected during a blood meal from an infected human host. The virus replicates in the mosquito’s gut and is subsequently transmitted to the salivary glands [4,5]. During a subsequent blood meal, the infected mosquito inoculates the virus into the new host [6]. While research has shown that mosquito salivary gland extracts can enhance viral infection in mammalian cells, the precise mechanisms underlying this process remain unclear [7].

Extracellular vesicles (EVs) are nanosized particles released by cells that play a crucial role in cell-to-cell communication (50–200 nm in diameter) [8]. When cells are activated or undergo differentiation, they release EVs into the extracellular space [9]. These EVs contain a diverse range of molecules, including proteins and metabolites, which can influence various cellular processes [10,11]. In the context of Zika virus infection, EVs may play a significant role in viral pathogenesis and host immune response. While the field of EV research is rapidly advancing, there is still much to learn about the specific mechanisms by which EVs function and their potential as biomarkers and therapeutic [12,13,14,15,16].

Omics analysis of isolated EVs is a rapidly evolving field with significant implications for various scientific disciplines. By characterizing the molecular composition of EVs, researchers can gain insights into cellular communication, disease pathogenesis, and potential therapeutic targets. In this study, we employed established metabolomic and proteomic techniques to comprehensively analyze EV samples. While this approach presents certain challenges, it has proven effective in generating valuable data and advancing our understanding of EV biology.

## 2. Materials and Methods

### 2.1. Ethical Aspects

The research adhered to the criteria outlined in the Helsinki conference. Before commencement, this study underwent scrutiny and gained approval from the Research Ethics Committee at the Federal University of Maranhão Hospital and Medicine School of Jundiaí (CAAE: 86696618.7.0000.5467). All actions followed the acknowledgment and endorsement expressed through the informed consent form.

### 2.2. Study Design

This research involved a prospective cohort of children residing in the State of Maranhão, located in northeastern Brazil. It was a collaborative effort with the Reference Center for Neurodevelopment, Assistance, and Rehabilitation of Children (NINAR). Additional data for control children in this study were primarily collected at the Infectious Pediatric Laboratory of the Medicine School of Jundiaí in Jundiaí. A blood sample (5 mL) was obtained from each child. Microcephaly, in this study, is defined as a head circumference measurement of less than two standard deviations (SDs) below the average [17,18]. Our study included a total of 14 children with confirmed Congenital Zika Syndrome (CZS+) and 15 children without Congenital Zika Syndrome (CZS-). Serums were fractionated by Izon qEV size-exclusion SEC, resulting in five fractions of 500 µL. The F3 fractions from 14 children with confirmed Congenital Zika Syndrome (CZS+) and 15 children without Congenital Zika Syndrome (CZS-) were analyzed using mass spectrometry techniques and metabolomic analysis. We subsequently carried out the characterization of isolated EV fractions through nanoparticle tracking analysis (NTA), and transmission electron microscopy (TEM) was also used. The protocols for isolation, characterization, and spectrometry analyses adhered to the methodology outlined by Pontes et al., 2020 [19].

### 2.3. Vesicle Isolation and Purification

Serum samples (500 μL) were processed through Izon qEV size-exclusion chromatography columns pre-conditioned with PBS containing 0.05% sodium azide. The columns, with a 75 nm pore size, 10 mL bed volume, and 3 mL void volume, were used as per the manufacturer’s instructions. After discarding the void volume, three 500 μL fractions were collected per sample [20,21].

### 2.4. EV Sample Preparation for MS-Based Proteomics Analysis

Extracellular vesicles (EVs) (F3) were stored in PBS with protease inhibitors at −80 °C for six months. They were then reconstituted in a lysis buffer containing urea, thiourea, DTT, and Triton X-100 for 2 h at room temperature. The buffer was removed using Amicon^®^ Ultra filters (Merck Millipore, Carrigtwohill, CO, USA), and protein quantification was performed using a NanoDrop One (ThermoFisher Scientific, Waltham, MA, USA). Protein banding patterns were analyzed by SDS-PAGE with BSA as a standard, using a 10% gel run at 50 V for 4.5 h. The gel was fixed in methanol/acetic acid solution, washed, and stained with GelCode. For EV protein digestion, the samples were DTT and alkylated with iodoacetamide in the dark. Digestion with trypsin occurred overnight, followed by acidification with formic acid to stop the reaction. The peptides were lyophilized, resuspended in TFA, and desalted using ZipTipC18 pipette tips (Merck Millipore, Carrigtwohill, CO, USA).

The processing parameters included the fixed carbamidomethylation of cysteine and variable modifications like methionine oxidation and N-terminal acetylation. Trypsin was used as the enzyme, allowing up to two cleavage errors. Mass tolerances were set at 20 ppm for peptides and 0.05 Da for fragments. Protein identification required a false discovery rate (FDR) of ≤1% and at least one peptide for confirmation. Adapted from the protocol Pontes et al., 2020 [19]

The bioinformatic analysis relied on information from UniProt *Cytoscape* and *String* (https://cytoscape.org/ and https://string-db.org/). The mass spectrometry proteomics data can be found in Appendix A—raw proteomics data.

### 2.5. EV Identification of Metabolites with GC-MS

The procedure involved the suspension of extracellular vesicles (EV) in PBS, following the methodology of Hoffman et al. with adjustments. Extraction was carried out in microtubes using 200 mg of fungal macerate and 1 mL of an ice-cold methanol/chloroform/water solution (6:2:2). After vigorous vortexing and an ultrasonic bath (20 Hz, 15 min), the tubes were centrifuged at 4 °C for 10 min at 14,000 rpm. The supernatant was filtered (0.22 μm filter) and lyophilized. The dried samples were reconstituted in 200 μL of extraction solution and analyzed by GC-MS and LC-MS/MS [22,23,24,25].

In LC-MS/MS, separation was performed on a reversed-phase C18 column using a linear gradient of mobile phase A (water with 0.1% formic acid) and mobile phase B (acetonitrile with 0.1% formic acid) over 10 min at a flow rate of 0.2 mL/min. The analytes were ionized using electrospray ionization (ESI) in positive mode and detected in multiple reaction monitoring (MRM) mode. The data were acquired using a triple quadrupole mass spectrometer operating in MRM mode, with dwell times of 100 ms for each transition. The data were processed using Analyst software, which included peak detection, integration, and quantification using external standards.

In GC-MS, the data were processed using ChromaTOF 4.32 software, with baseline correction, deconvolution, retention indices (RIs), retention times (RTs) and peak alignment. Identification used the NIST library (metabolites with a score ≥700), and intensity was normalized by the total ion content (TIC). The methodology was based on Budzinski et al.’s work, with adjustments, including the use of n-alkanes (C12-C40) for IR calculation. Automated injection was carried out in splitless mode on an Agilent 7890A chromatograph.

Proteomic data analysis was performed using Progenesis QI for Proteomics v4.2 software. The raw data were processed by applying the following steps: peak detection, alignment, calibration, and global normalization using the quantile normalization method. Then, the data were logarithmically transformed (base 10). Peptide identification was performed via comparison with the Homo sapiens FASTA protein database obtained from UniProt on 15 January 2023. For quantification, the label-free method based on peak intensity was used. The statistical significance of differences between groups was assessed using Student’s *t*-test with Benjamini–Hochberg correction to control the false positive rate. Differentially expressed proteins were subjected to functional enrichment analysis using the PANTHER system software (Hologic Inc., Marlborough, MA, USA).

## 3. Results

### 3.1. EV Characterization

In molecular exclusion liquid chromatography, four fractions containing extracellular vesicles were obtained. Although all of the fractions contain vesicles, according to the manufacturer’s information, fraction 3 exhibits the highest purity, making it suitable for further analyses. Densitometry analysis of the Western blot bands of Fractions F3 and F4 in triplicate can be seen in Figure 1. For this analysis, the area of the opted bands was analysed using Image J and compared with each other to better visualise the yield of Fraction 3.

To isolate EVs, the collected sera underwent fractionation using a molecular exclusion chromatography column, and the resulting isolates were preserved. Subsequently, quantification, nanoparticle tracking, and transmission electron microscopy analysis were performed on the isolates. The sizes of the isolated extracellular vesicles ranged from 154.2 nm (minimum) to 196.4 nm (maximum) for the control child pool (CZS-) and from 175.0 nm (minimum) to 196.4 nm (maximum) for the affected child pool (CZS+), with an average size of 154.9 nm for the CZS- group and 185.4. nm for the CZS+ group. The polydispersity index (PDI) ranged from 0.17 to 0.20 for CZS- and 0.19 to 0.24 for CZS+. The concentration of particles was between 3.4 × 10^8^ particles/mL (minimum) and 4.2 × 10^8^ particles/mL (maximum). Quantification using NanoDrop analysis showed values ranging from 36.5 ng/μL to 80.4 ng/μL (Table 1) [26,27,28,29].

### 3.2. Proteomic and Metabolomic Analysis

Leveraging its Cytoscape plugin, we characterized functionality and generated a protein–metabolite network, revealing novel insights into pathways like the negative regulation of lipid localization and complement activation, an alternative pathway compared to the control. The aim is to identify (1) biomarkers: proteins and metabolites that correlate strongly with immune status and can be used to diagnose or monitor diseases; (2) key metabolic pathways: networks of metabolic interactions that are modulated during the immune response; and (3) potential therapeutic targets: proteins and metabolites that can be manipulated to modulate the immune response. In summary, this correlation network provides a comprehensive view of the molecular interactions that occur during the immune response, allowing for the identification of new therapeutic targets and the understanding of the molecular mechanisms underlying various diseases.

The figure shows a correlation network that visualizes the interactions between proteins and metabolites in the context of the immune response. The analysis was carried out using proteomic and metabolomic data, and the visualization was generated using Cytoscape 3 software. Each node represents the following: protein, which is represented by a blue circle, and metabolites, which are represented by a yellow square. The edges (lines) connect the following: pairs of proteins or metabolites, indicating a significant correlation between them, and proteins and metabolites, suggesting a possible functional interaction. The thickness of the edges reflects the strength of the correlation, with thicker lines indicating stronger correlations. The colour of the edges indicates the type of correlation, namely red, which denotes a positive correlation (when one increases, the other also increases), and blue, which denotes a negative correlation (when one increases, the other decreases).

Figure 2 and Figure 3 present a comprehensive correlation network visualizing the intricate interplay between proteins and metabolites within the immune response. Leveraging proteomic and metabolomic data analyzed and visualized using Cytoscape 3, these networks offer valuable insights into the molecular mechanisms underlying immune function. The nodes represent proteins (blue circles) and metabolites (yellow squares), while edges (lines) signify significant correlations or potential functional interactions. Edge thickness reflects correlation strength, and colour indicates correlation type: red for positive and blue for negative. By identifying key biomarkers, metabolic pathways, and potential therapeutic targets, these networks contribute to a deeper understanding of the immune response and may facilitate the development of novel therapeutic strategies for various immune-related diseases.

## 4. Discussion

Studies on viral infections such as DENV and H5N1 have explored the inhibition of ACE and kinases involved in the PI3K-AKT-mTOR pathway. Data from the literature suggest that the cellular location of certain functions is as follows: (1) cytoplasm: the regulation of cell growth, proliferation, and survival; (2) nucleus: the transcriptional regulation of genes involved in cell cycle progression and apoptosis; (3) mitochondria: the regulation of mitochondrial bioenergetics and oxidative stress; and (4) plasma membrane: signalling pathways involved in cell adhesion, migration, and immune response. These findings provide valuable insights into the molecular mechanisms underlying the pathogenesis of Zika and suggest potential therapeutic targets for the development of novel treatments of the current and observed inflammation process [10,30,31]. Considering this, we can highlight the presence of upregulated EV metabolites, predominantly involved in the PI3K-AKT-mTOR pathway. These findings suggest a significant role in the inhibition of Angiotensin in children with CZS+. Studies on viral infections such as DENV and H5N1 have explored the inhibition of ACE [32,33,34,35] and kinases involved in the PI3K-AKT-mTOR pathway [36,37]. Data from the literature suggest that cellular locations may determine certain functions [38,39,40].

ZIKV infection during pregnancy carries a high risk of severe birth defects, including microcephaly, a condition characterized by an abnormally small head size [38,41]. Neurological complications such as encephalopathies, meningoencephalitis, myelitis, uveitis, Guillain–Barré syndrome, and severe thrombocytopenia have also been linked to ZIKV infection [42,43,44]. The precise mechanisms underlying these severe forms of ZIKV disease remain unclear [44]. Currently, there are no effective vaccines or specific treatments available for ZIKV [45,46,47]. The increasing prevalence and spread of ZIKV are attributed to the virulence of circulating strains, the vulnerability of populations, and the efficient transmission by Aedes aegypti and Aedes albopictus mosquitoes. These mosquitoes transmit various Flaviviruses, including ZIKV and dengue (DENV), by feeding on infected hosts and transferring the virus to their salivary glands [48,49]. The subsequent feeding on a new host completes the transmission cycle. Researchers have observed an increase in mammalian cell infection following exposure to mosquito salivary gland extracts, particularly in DENV infections. However, the specific mechanisms involved in Flavivirus transmission from vectors to hosts are not fully understood [50,51].

Extracellular vesicles (EVs) are cell-secreted, membrane-enclosed particles ranging from 50 to 200 nm in diameter. In humans infected with ZIKV, the primary target cells include monocytes, macrophages, endothelial cells, and neurons. When cells differentiate or become activated, they release EVs, also known as exosomes or microvesicles, into the extracellular space [52,53,54]. EVs contain metabolites and proteins that play crucial roles in various biological processes. Studying the metabolomes and proteomes of EVs is becoming increasingly important for understanding cellular behaviour, similar to how EVs are used to investigate dysregulations. Notably, EVs exhibit metabolic and proteolytic activity, as evidenced by metabolomic analysis of serum samples alongside hepatocyte EVs. While the field of EV metabolomics and proteomics holds great promise, further research is needed to fully characterize the diverse classes of EVs and their inherent diversity. Existing studies in the literature suggest that PI3K inhibitors targeting the stroma and focusing on immune modulation might be a promising therapeutic avenue [55].

EVs play a critical role in cellular communication, influencing various physiological and pathological processes, including immune responses, cancer development, cardiovascular diseases, and the delivery of bacterial virulence factors to the host [56,57,58]. They also participate in modulating inflammatory and immune responses during fetal–maternal communication. EVs facilitate communication between different cell types, particularly immune system cells and their secreted molecules. Their immunomodulatory capacity has gained significant attention in recent years [59]. EVs act as vehicles for transferring membrane, cytosolic proteins, lipids, and other molecules between cells, suggesting their role as communicators in cellular interactions. When secreted by immune system cells, EVs exhibit immunomodulatory functions, inducing both the suppression and activation of immune responses [60,61,62,63,64,65].

Interestingly, our findings align with previous observations in patients with increased interferon alpha (IFN-α) production by plasmacytoid dendritic cells. This was observed through their interaction with EVs originating from a hepatocarcinoma cell line infected with the hepatitis C virus (HCV). This suggests a potential defence strategy employed by the organism to activate the innate immune response against viral infection. Additionally, several proteins identified in this study as potential biomarkers for serum EVs in children with CZS+ have been previously described in the context of HIV-1 [66].

The results presented in this study provide new evidence on the molecular mechanisms underlying Zika virus pathogenesis and the role of extracellular vesicles (EVs) in this process. The identification of specific proteins and metabolites in EVs isolated from children with Congenital Zika Syndrome (CZS) suggests that these particles may serve as important biomarkers and therapeutic targets. One of the most significant findings of this study is the association between Zika virus infection and the inhibition of the PI3K-AKT-mTOR pathway. This pathway plays a crucial role in neuronal development, and its inhibition may contribute to the development of microcephaly. The identification of metabolites that modulate this pathway in EVs isolated from patients with CZS suggests that manipulating this pathway could be a promising therapeutic strategy [67].

## 5. Conclusions

The findings of this study have important clinical implications. The identification of specific biomarkers in EVs may allow for an earlier and more accurate diagnosis of CZS; it may also help monitor disease progression and evaluate the response to treatment. In addition, understanding the molecular mechanisms underlying the pathogenesis of the Zika virus may lead to the development of new therapies to prevent and treat this disease. The ability of EVs to transport proteins, lipids, and nucleic acids between cells makes them potent modulators of immune response, angiogenesis, and tumour metastasis. In the context of Zika virus infection, EVs can play a dual role. On one hand, they can serve as a viral dissemination mechanism, facilitating the infection of new cells. On the other hand, they can induce a robust immune response, contributing to the elimination of the virus.

## Figures and Tables

**Figure 1 biomolecules-15-00032-f001:**
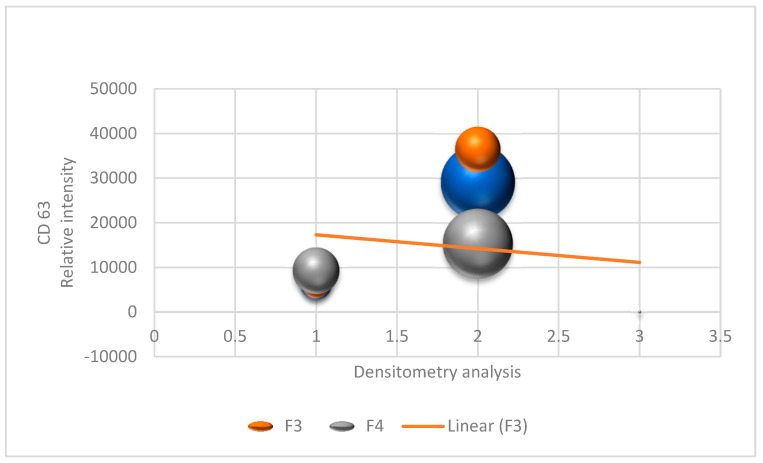
Densitometry analysis of Western blot bands for Fractions F3 and F4.

**Figure 2 biomolecules-15-00032-f002:**
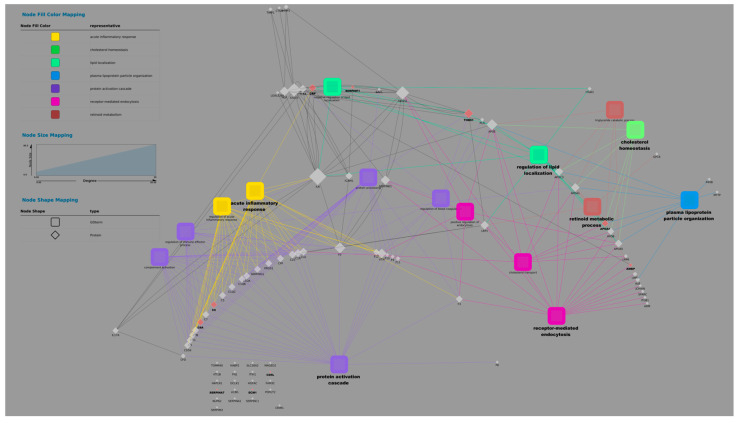
Proteomic and metabolomic correlation network of the immune system.

**Figure 3 biomolecules-15-00032-f003:**
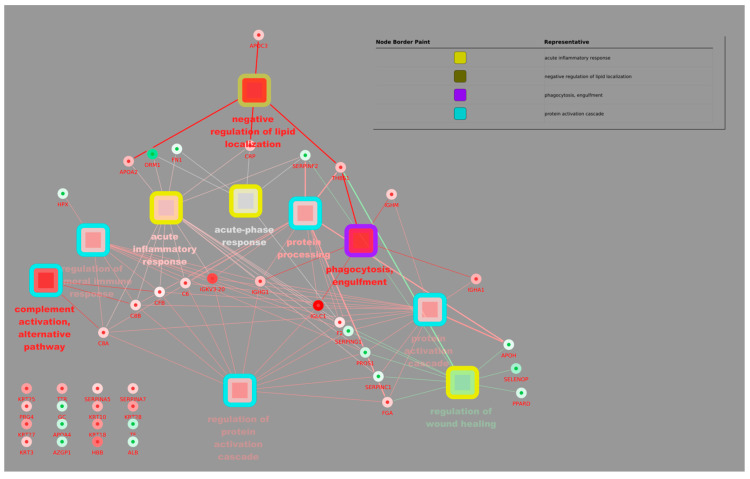
Proteomic and metabolomic correlation network of the immune system. The figure shows a correlation network that visualizes the interactions between proteins and metabolites in the context of the immune response. The analysis was carried out using proteomic and metabolomic data, and the visualization was generated using Cytoscape 3 software. Each node represents the following: protein, represented by a blue circle, and metabolites, represented by a yellow square. The edges (lines) connect the following: pairs of proteins or metabolites, indicating a significant correlation between them, and proteins and metabolites, suggesting a possible functional interaction. The thickness of the edges reflects the strength of the correlation, with thicker lines indicating stronger correlations. The colour of the edges indicates the type of correlation, namely red, indicating positive correlation (when one increases, the other also increases), and blue, indicating negative correlation (when one increases, the other decreases).

**Table 1 biomolecules-15-00032-t001:** Characterization of extracellular vesicles (EVs) isolated from serum of children with and without Congenital Zika Syndrome (CZS).

Sample	Group	Average Size (nm)	Minimum Size (nm)	Maximum Size (nm)	Dispersion (PDI)	Concentration (Particles/mL)	Quantification (ng/ µL)
**CZS +1**	CZS+	175.0	140	220	0.22	3.8 × 10 ^8^	52.3
**CZS +2**	CZS+	190.2	135	225	0.20	4.1 × 10 ^8^	60.1
**CZS +3**	CZS+	180.0	150	230	0.23	4.0 × 10 ^8^	45.0
**CZS +4**	CZS+	195.3	140	210	0.19	3.9 × 10 ^8^	70.5
**CZS +5**	CZS+	178.5	145	220	0.21	3.7 × 10 ^8^	50.7
**CZS +6**	CZS+	192.6	160	235	0.24	4.2 × 10 ^8^	80.4
**CZS +7**	CZS+	170.0	150	225	0.21	4.0 × 10 ^8^	55.3
**CZS +8**	CZS+	196.4	130	220	0.22	3.9 × 10 ^8^	65.2
**CZS +9**	CZS+	188.9	145	225	0.20	4.1 × 10 ^8^	61.7
**CZS +10**	CZS+	179.7	135	215	0.22	4.0 × 10 ^8^	48.6
**CZS +11**	CZS+	193.8	140	220	0.19	3.8 × 10 ^8^	72.4
**CZS +12**	CZS+	180.5	150	230	0.21	4.1 × 10 ^8^	58.9
**CZS +13**	CZS+	194.2	140	220	0.22	3.9 × 10 ^8^	69.5
**CZS +14**	CZS+	185.4	130	220	0.21	4.0 × 10 ^8^	57.2
**CZS -1**	CZS-	150.0	120	200	0.19	3.5 × 10 ^8^	44.1
**CZS -2**	CZS-	160.1	115	195	0.18	3.4 × 10 ^8^	36.5
**CZS -3**	CZS-	152.0	110	200	0.18	3.5 × 10 ^8^	38.9
**CZS -4**	CZS-	159.5	125	205	0.19	3.6 × 10 ^8^	42.6
**CZS -5**	CZS-	149.0	130	210	0.20	3.7 × 10 ^8^	49.2
**CZS -6**	CZS-	160.30	120	200	0.18	3.5 × 10 ^8^	46.7
**CZS -7**	CZS-	155.5	125	205	0.19	3.6 × 10 ^8^	43.8
**CZS -8**	CZS-	153.8	110	195	0.17	3.4 × 10 ^8^	37.6
**CZS -9**	CZS-	148.9	120	200	0.18	3.5 × 10 ^8g^	41.4
**CZS -10**	CZS-	161.0	130	210	0.20	3.6 × 10 ^8^	49.8
**CZS -11**	CZS-	156.7	115	195	0.18	3.5 × 10 ^8^	44.9
**CZS -12**	CZS-	151.2	120	200	0.17	3.4 × 10 ^8^	39.2
**CZS -13**	CZS-	157.0	125	205	0.19	3.6 × 10 ^8^	45.3
**CZS -14**	CZS-	153.4	120	200	0.18	3.5 × 10 ^8^	42.0
**CZS -15**	CZS-	154.9	120	198	0.20	3.6 × 10 ^8^	40.5

## Data Availability

The datasets generated by this study will be available upon reasonable request to the authors.

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
