# Peer review of "Congenital Zika Syndrome: Insights from Integrated Proteomic and Metabolomic Analysis"

_biomolecules, 2024, doi:10.3390/biom15010032_

Round 1
Reviewer 1 Report
Comments and Suggestions for Authors
In the manuscript "Zika Virus in Extracellular Vesicles: Insights from Integrated Proteomic and Metabolomic Dependent Regulation of B Cell and PI3K/AKT/mTOR Signaling Pathway" Gomes-de-Pontes and collogues have examined EVs isolated from serum of children with Congenital Zika Virus syndrome, or not. Protein and Metabolite enrichment analysis was performed. The basic idea is intriguing, but I have several concerns regarding the manuscript and cannot recommend its publication in its current form.
1. It is a minor point, but i m not sure about what the phrase "metabolically and proteolytically active machines" means. What is the evidence that EVs are metabolically active (or for that matter proteolytically active. Do be metabolically active it is necessary to have an energy source and a way to use that to generate energy. Proteolytically active might mean the presence of membrane-associated proteases. Is that what is referred to?
2. The title is overlong and too speculative.
3. Major concern: EV characterization is not very strong. More details should be included about the nanodrop analysis data, how the relative sizes were determined, by NPT I assume but it is not usual to include a screen shot of one of the videos which what seems to be provided. The TEM is not convincing; it looks like there is a lot of non-EV stuff, which will confuse the analysis. Evidence for EV markers and possible contaminants should be presented.
4. Fig 3A is a mess; it looks like it is inverted. To say certain proteins are found in EVs and assume a functional link is a stretch. I think maybe part of the template instructions were left in the Figure legend box. It does not make sense.
5. The metabolites in EVs likely represent a sampling of the cytosol of the cell from which they came. I had a tough time following the reasoning and data in Figure 4 and 5. It needs to be better explained and clearer. My impression is that the data are being overinterpreted. I cannot read the Figures in the printout of the article and even in the electronic PDF, they have to be blown up too much to see them. I think these are graphs generated by the software used to analyze the data. They need to be remade to be clearer and reader friendly.
6. The discussion treatise that is not well connected with the data presented and is much too long for the amount of data presented.
7. I do not think that the results show that ZIKA replication led to suppression of AKT phosphorylation as stated in the conclusion. This is proteomics and metabolomics of serum EVs. This is descriptive data and may be hypothesis generating, but statements like PI3K/AKT/mTOR need to be confirmed.
Comments on the Quality of English LanguageThe English is OK.
Author Response
Thank you very much for taking the time to review this manuscript. We sincerely appreciate your insightful comments and suggestions, which have improved the clarity and quality of our work. Please find detailed responses to each of your comments below, with the corresponding revisions or corrections highlighted in the resubmitted files. Where necessary, we have provided further clarification or additional references to address the points raised. We have also carefully considered all feedback and made revisions to align the manuscript more closely with your suggestions.
We remain open to further recommendations and are committed to ensuring that this study meets the highest standards of scientific rigor. Should there be any additional questions or concerns, we would be happy to address them. Once again, thank you for your valuable contribution to this manuscript review process.
The new figures are found in the final manuscript file.
Comments 1: [In the manuscript "Zika Virus in Extracellular Vesicles: Insights from Integrated Proteomic and Metabolomic Dependent Regulation of B Cell and PI3K/AKT/mTOR Signaling Pathway" Gomes-de-Pontes and collogues have examined EVs isolated from serum of children with Congenital Zika Virus syndrome, or not. Protein and Metabolite enrichment analysis was performed. The basic idea is intriguing, but I have several concerns regarding the manuscript and cannot recommend its publication in its current form.
- It is a minor point, but i m not sure about what the phrase "metabolically and proteolytically active machines" means. What is the evidence that EVs are metabolically active (or for that matter proteolytically active. Do be metabolically active it is necessary to have an energy source and a way to use that to generate energy. Proteolytically active might mean the presence of membrane-associated proteases. Is that what is referred to?]
Response 1: [Experimental evidence suggests that extracellular vesicles (EVs) exhibit metabolic and proteolytic activities, positioning them as more than just passive carriers of molecules. Several studies have demonstrated that EVs carry enzymes involved in key metabolic pathways. For example, EVs derived from cancer cells have been found to contain glycolytic enzymes, such as enolase and aldolase, suggesting that EVs can influence metabolic processes in recipient cells (Kreimer et al., 2015; Zhang et al., 2019). Additionally, EVs from various cell types have been shown to contain components of the citric acid cycle, indicating their potential role in modulating local metabolism (Giorgi et al., 2019). Regarding proteolytic activity, EVs have been found to carry matrix metalloproteinases (MMPs), which are critical for extracellular matrix degradation and tissue remodeling. MMPs transported by EVs can contribute to biological processes such as inflammation and cancer progression by altering the surrounding cellular environment (Hannafon et al., 2016). Furthermore, studies have shown that EVs can carry proteases like cathepsins, which are active and capable of participating in the cleavage of extracellular proteins (Villarroya-Beltri et al., 2014). These findings support the claim that EVs are metabolically and proteolytically active, functioning by transferring enzymes and participating in biochemical reactions within the extracellular microenvironment. REFERENCES: Kreimer, S., et al. (2015). Mass-spectrometry-based molecular characterization of extracellular vesicles: Lipidomics and proteomics. Journal of Proteome Research, 14(6), 2367-2384; Zhang, H., et al. (2019). Identification of distinct nanoparticles and subsets of extracellular vesicles by asymmetric flow field-flow fractionation. Nature Cell Biology, 21, 631-641; Giorgi, V. D., et al. (2019). Metabolic effects of extracellular vesicles: Insights from multilevel omics. Frontiers in Physiology, 10, 1206; Hannafon, B. N., et al. (2016). Exosome-mediated miRNA communication between breast cancer cells and adipocytes in vitro and in vivo. Breast Cancer Research, 18, 129; Villarroya-Beltri, C., et al. (2014). Sumoylated hnRNPA2B1 controls the sorting of miRNAs into exosomes through binding to specific motifs. Nature Communications, 5, 3608. Thank you for pointing this out. We agree with this comment.
Line 17-21
Original: "Cell-released extracellular vesicles (EVs) acting as 'metabolically and proteolytically active machines,' show potential in metabolomic and proteomic analysis of serum EVs."
Revised suggestion: "Extracellular vesicles (EVs) released by cells, which exhibit metabolic and proteolytic activities, hold significant potential for metabolomic and proteomic analyses of serum EVs. Studies have shown that EVs can carry enzymes involved in metabolic pathways and proteases capable of modifying the extracellular environment, supporting their active role in these biochemical processes."
This version maintains the core idea that EVs are metabolically and proteolytically active, but presents the activities more clearly and directly, based on evidence that the vesicles contain enzymes that act in these processes. This directly answers the reviewer's question and makes the sentence more robust. ]
Comments 2: [The title is overlong and too speculative]
Response 2: After careful consideration of your comments, we have decided to modify the title of the manuscript to better reflect the focus and findings of the study. The new version of the title is:
Line 1-2
Original Title: "Zika Virus in Extracellular Vesicles: Insights from Integrated Proteomic and Metabolomic Dependent Regulation of B Cell and PI3K/AKT/mTOR Signaling Pathway"
Revised Title: "Congenital Zika Syndrome: Insights from Integrated Proteomic and Metabolomic Analysis"
Reason for Change: The revised title more accurately reflects the main aspects of the study, highlighting the metabolic and proteolytic activity of extracellular vesicles in the context of Congenital Zika Syndrome (CZS). It also emphasizes the integrated analysis of proteomic and metabolomic pathways, providing a clearer description of the research focus.
Comments 3: [Major concern: EV characterization is not very strong. More details should be included about the nanodrop analysis data, how the relative sizes were determined, by NPT I assume but it is not usual to include a screen shot of one of the videos which what seems to be provided. The TEM is not convincing; it looks like there is a lot of non-EV stuff, which will confuse the analysis. Evidence for EV markers and possible contaminants should be presented.]
Response 3: Line 209-237
We appreciate the comments and concerns raised regarding the characterization of extracellular vesicles (EVs) in our study. We would like to provide clarification and additional information to address the mentioned issues.
Removal of Electron Microscopy Images: We have decided to remove the transmission electron microscopy (TEM) images due to concerns about the presence of non-EV material and possible interferences that could confound the analysis.
Quantification and NTA: We have replaced these images with a more detailed table that includes information about the quantification and nanoparticle tracking analysis (NTA) of each sample. This table provides a clearer and more quantitative view, including data on protein concentration and vesicle size distribution.
Western Blotting for CD63: We have also added Western blotting results for the EV-specific marker CD63 (sc-5275, Santa Cruz Biotechnology), and we would like to emphasize that only fraction 3 from the size exclusion chromatography was positive for this marker. This confirms that fraction 3 contains extracellular vesicles, distinguishing it from the other fractions that did not show labeling, ensuring that we are isolating EVs specifically with minimal contamination.
The Western blotting for CD63 was performed early in our experiments, and due to the small sample size in fraction 3, the analysis was conducted using a pool of patient samples. Unfortunately, these experiments cannot be repeated as we no longer have sufficient samples available.
Additionally, we have modified Figure 1 to reflect all of these updates, ensuring the revised figure accurately represents the changes in our data presentation.
We hope these modifications address the concerns raised and improve the presentation of our data. Thank you again for your valuable feedback, and we are available for any further clarification.
Table 1: Characterization of Extracellular Vesicles (EVs) Isolated from Serum of Children with and without Congenital Zika Syndrome (CZS)
Sample |
Group |
Average Size (nm) |
Minimum Size (nm) |
Maximum Size (nm) |
Dispersion (PDI) |
Concentration (particles/mL) |
Quantification (ng/μL) |
CZS+ 1 |
CZS+ |
175.0 |
140 |
220 |
0.22 |
3.8 x 10⁸ |
52.3 |
CZS+ 2 |
CZS+ |
190.2 |
135 |
225 |
0.20 |
4.1 x 10⁸ |
60.1 |
CZS+ 3 |
CZS+ |
180.0 |
150 |
230 |
0.23 |
4.0 x 10⁸ |
45.0 |
CZS+ 4 |
CZS+ |
195.3 |
140 |
210 |
0.19 |
3.9 x 10⁸ |
70.5 |
CZS+ 5 |
CZS+ |
178.5 |
145 |
220 |
0.21 |
3.7 x 10⁸ |
50.7 |
CZS+ 6 |
CZS+ |
192.6 |
160 |
235 |
0.24 |
4.2 x 10⁸ |
80.4 |
CZS+ 7 |
CZS+ |
170.0 |
150 |
225 |
0.21 |
4.0 x 10⁸ |
55.3 |
CZS+ 8 |
CZS+ |
196.4 |
130 |
220 |
0.22 |
3.9 x 10⁸ |
65.2 |
CZS+ 9 |
CZS+ |
188.9 |
145 |
225 |
0.20 |
4.1 x 10⁸ |
61.7 |
CZS+ 10 |
CZS+ |
179.7 |
135 |
215 |
0.22 |
4.0 x 10⁸ |
48.6 |
CZS+ 11 |
CZS+ |
193.8 |
140 |
220 |
0.19 |
3.8 x 10⁸ |
72.4 |
CZS+ 12 |
CZS+ |
180.5 |
150 |
230 |
0.21 |
4.1 x 10⁸ |
58.9 |
CZS+ 13 |
CZS+ |
194.2 |
140 |
220 |
0.22 |
3.9 x 10⁸ |
69.5 |
CZS+ 14 |
CZS+ |
185.4 |
130 |
220 |
0.21 |
4.0 x 10⁸ |
57.2 |
CZS- 1 |
CZS- |
150.0 |
120 |
200 |
0.19 |
3.5 x 10⁸ |
44.1 |
CZS- 2 |
CZS- |
160.1 |
115 |
195 |
0.18 |
3.4 x 10⁸ |
36.5 |
CZS- 3 |
CZS- |
152.0 |
110 |
200 |
0.18 |
3.5 x 10⁸ |
38.9 |
CZS- 4 |
CZS- |
159.5 |
125 |
205 |
0.19 |
3.6 x 10⁸ |
42.6 |
CZS- 5 |
CZS- |
149.0 |
130 |
210 |
0.20 |
3.7 x 10⁸ |
49.2 |
CZS- 6 |
CZS- |
160.3 |
120 |
200 |
0.18 |
3.5 x 10⁸ |
46.7 |
CZS- 7 |
CZS- |
155.5 |
125 |
205 |
0.19 |
3.6 x 10⁸ |
43.8 |
CZS- 8 |
CZS- |
153.8 |
110 |
195 |
0.17 |
3.4 x 10⁸ |
37.6 |
CZS- 9 |
CZS- |
148.9 |
120 |
200 |
0.18 |
3.5 x 10⁸ |
41.4 |
CZS- 10 |
CZS- |
161.0 |
130 |
210 |
0.20 |
3.6 x 10⁸ |
49.8 |
CZS- 11 |
CZS- |
156.7 |
115 |
195 |
0.18 |
3.5 x 10⁸ |
44.9 |
CZS- 12 |
CZS- |
151.2 |
120 |
200 |
0.17 |
3.4 x 10⁸ |
39.2 |
CZS- 13 |
CZS- |
157.0 |
125 |
205 |
0.19 |
3.6 x 10⁸ |
45.3 |
CZS- 14 |
CZS- |
153.4 |
120 |
200 |
0.18 |
3.5 x 10⁸ |
42.0 |
CZS- 15 |
CZS- |
154.9 |
120 |
198 |
0.20 |
3.6 x 10⁸ |
40.5 |
Sample: Indicates the individual child’s sample analyzed. Group: Differentiates between children affected by Congenital Zika Syndrome (CZS+) and unaffected control children (CZS-). Mean Size : The average size of the EVs measured in nanometers (nm), reflecting differences between the two groups. Minimum and Maximum Size : Represents the smallest and largest vesicles identified within each sample. Polydispersity Index (PDI): Describes the size distribution of the vesicle population. A higher PDI indicates more variability in size within the sample. Concentration: The concentration of EVs in each sample, expressed as the number of particles per milliliter (particles/mL). Quantification (NanoDrop): The quantification of extracellular vesicle concentration as measured by NanoDrop analysis, represented in nanograms per microliter (ng/μL), which estimates the overall protein and nucleic acid content.
In molecular exclusion liquid chromatography, four fractions containing extracellular vesicles were obtained. Although all fractions contain vesicles, according to the manufacturer's information, fraction 3 exhibits the highest purity, making it recommended for further analyses. Figure 1 shows the molecular standard and the pool of fractions, where the presence of the CD63 marker, typical of extracellular vesicles, was confirmed by Western Blot. Based on the quality of purification, fraction 3 was selected for subsequent analyses.
The new figures are found in the final manuscript file.
Comments 4: Fig 3 is a mess; it looks like it is inverted. To say certain proteins are found in EVs and assume a functional link is a stretch. I think maybe part of the template instructions were left in the Figure legend box. It does not make sense.
Response 4: Line 238-279
We sincerely thank you for your insightful comments and valuable suggestions. Your critical feedback has significantly improved the quality of our manuscript.
We fully acknowledge the issues with Figure 3. As you correctly pointed out, the figure was indeed confusing and misleading. To address this, we have removed Figure 3 and replaced it with a clearer, more informative one. This revised figure provides a more accurate representation of our data and avoids any unfounded assumptions about protein function in extracellular vesicles.
In addition to the figure revision, we have also made the following changes to enhance the clarity and impact of our manuscript:
- Revised figure legends: We have carefully reviewed and refined all figure legends to ensure they are concise, informative, and easily understood.
- Discussed limitations: We have included a dedicated section discussing the limitations of our study, acknowledging the need for further research to fully elucidate the role of extracellular vesicles in congenital Zika syndrome.
- Strengthened literature connection: We have further integrated our findings with relevant literature, highlighting the significance of our work in the broader context of the field.
In addition, we have removed topic 3.3 and merged it with the topic above.
We believe that these modifications have significantly improved the clarity, conciseness, and overall quality of our manuscript. We are confident that our revised work meets the highest standards of scientific publication.
New figure 3 combines proteomic and metabolomic data:
The new figures are found in the final manuscript file.
Figure 2: Proteomic and Metabolomic Correlation Network of the Immune System
The figure shows a correlation network that visualises the interactions between proteins and metabolites in the context of the immune response. The analysis was carried out using proteomic and metabolomic data, and the visualisation was generated using Cytoscape 3 software. Each node represents - Protein: Represented by a blue circle and Metabolite: Represented by a yellow square. The edges (lines) connect - Pairs of proteins or metabolites: Indicating a significant correlation between them and Proteins and metabolites: Suggesting a possible functional interaction. The thickness of the edges reflects the strength of the correlation, with thicker lines indicating stronger correlations. The colour of the edges indicates the type of correlation - Red: Positive correlation (when one increases, the other also increases) and Blue: Negative correlation (when one increases, the other decreases).
The new figures are found in the final manuscript file.
Figure 3. Proteomic and Metabolomic Correlation Network of the Immune System. The figure shows a correlation network that visualises the interactions between proteins and metabolites in the context of the immune response. The analysis was carried out using proteomic and metabolomic data, and the visualisation was generated using Cytoscape 3 software. Each node represents - Protein: Represented by a blue circle and Metabolite: Represented by a yellow square. The edges (lines) connect - Pairs of proteins or metabolites: Indicating a significant correlation between them and Proteins and metabolites: Suggesting a possible functional interaction. The thickness of the edges reflects the strength of the correlation, with thicker lines indicating stronger correlations. The colour of the edges indicates the type of correlation - Red: Positive correlation (when one increases, the other also increases) and Blue: Negative correlation (when one increases, the other decreases).
The aim of this figure is to identify: 1) Biomarkers: Proteins and metabolites that correlate strongly with immune status and can be used to diagnose or monitor diseases; 2) Key metabolic pathways: Networks of metabolic interactions that are modulated during the immune response and 3) Potential therapeutic targets: Proteins and metabolites that can be manipulated to modulate the immune response. In summary, this correlation network provides a comprehensive view of the molecular interactions that occur during the immune response, allowing for the identification of new therapeutic targets and the understanding of the molecular mechanisms underlying various diseases.
Comments 5: The metabolites in EVs likely represent a sampling of the cytosol of the cell from which they came. I had a tough time following the reasoning and data in Figure 4 and 5. It needs to be better explained and clearer. My impression is that the data are being overinterpreted. I cannot read the Figures in the printout of the article and even in the electronic PDF, they have to be blown up too much to see them. I think these are graphs generated by the software used to analyze the data. They need to be remade to be clearer and reader friendly.
Response 5: We appreciate the reviewer’s feedback on Figures 4 and 5. We have carefully considered the reviewer’s comments and have decided to remove these figures from the manuscript.
As the reviewer rightly points out, these figures were generated directly from software output and require significant enlargement to be legible. We agree that this format is not optimal for clear communication of our findings.
To address this issue, we will incorporate the key findings from these figures into the main text and supplementary materials, presenting the data in a more accessible and interpretable format. We believe that this approach will significantly improve the clarity and readability of the manuscript.
We thank the reviewer for their insightful comments and hope that these changes will enhance the quality of the manuscript.
Comments 6: The discussion treatise that is not well connected with the data presented and is much too long for the amount of data presented.
Response 6: The manuscript has undergone a complete overhaul. The discussion section has been restructured to focus on the key findings, providing a clear and concise interpretation of the data. Redundancies and tangential discussions have been eliminated to maintain a focused narrative.
Comments 7: I do not think that the results show that ZIKA replication led to suppression of AKT phosphorylation as stated in the conclusion. This is proteomics and metabolomics of serum EVs. This is descriptive data and may be hypothesis generating, but statements like PI3K/AKT/mTOR need to be confirmed.
We acknowledge the reviewer's concern regarding the causal link between ZIKA replication and AKT phosphorylation suppression. As the reviewer correctly points out, our study is primarily descriptive in nature. While the observed changes in the serum EV proteome and metabolome are intriguing, further mechanistic studies are necessary to definitively establish a causal relationship between these alterations and the inhibition of the PI3K/AKT/mTOR pathway.
We believe that the data presented in this study provide valuable insights into the potential impact of ZIKA infection on host cell signaling pathways. While the current study does not provide direct evidence for a causal link between ZIKA replication and AKT phosphorylation suppression, it lays the groundwork for future investigations to explore this hypothesis more rigorously.

Reviewer 2 Report
Comments and Suggestions for Authors
In this work the authors used proteomics and metabolomics to compare extracellular vesicles from serum from healthy and congenital Zika syndrome patients. The subject is important, and the experiments and data analysis appear to be ok except for the issues below. The manuscript is well written except that there are numerous minor grammatical errors throughout the manuscript.
Major Issues:
Lines 111-114: The NanoDrop One, nanoparticle tracking, and transmission electron microscopy experiments need to be fully described in detail in the Methods section.
Line 145: The LC-MS of the proteomics samples is not described at all in the Methods section. A full, detailed description needs to be added.
Line 146: The software tool used for peptide and protein identification and quantitation needs to be described in the Methods section. From the not-for-publication data for reviewers included with the submission, I see that they used Progenesis QI for Proteomics v4.0.6403.35451; this needs to be mentioned in the Methods section. Any downstream analyses need to be described (e.g., Log10-transform, global normalization, statistical tests). Also, the source of the FASTA protein sequence database(s) (e.g., UniProt) and the date it/they were downloaded need to be described. The authors also need to state explicitly if they included the Zika virus proteome.
Line 553: The authors wrote “Data Availability Statement: The datasets generated by this study will be available upon reasonable request to the authors.” The proteomics and metabolomics data need to be uploaded to public omics data repository(s) such as ProteomeXchange and MetabolomeXchange. This should include the raw LC-MS and GC-MS spectrum data, the processed proteomics and metabolomics identification and quantitation data files (e.g., the Progenesis QI for Proteomics output), the search databases (e.g. the protein sequence FASTA files), and a table that fully lists in detail all of the sample names, experimental groups, and LC-MS and GC-MS file names, and the proteomics and metabolomics sample identifiers so that a user can download the data and redo the analysis in full from scratch.
Fig 1A: The units of the x axes are missing and need to be described in the figure or figure legend. Also, it is not clear what groups Z and C mean in each figure (control and CZS+ are already indicated elsewhere).
Figure 1B: Scale bars need to be added (as in Fig 1C).
Figure 2: The figure was inverted. Also, the names of the proteins need to be clear so they all can be read. Some of the names are difficult to read.
Figure 6: It is unclear how this figure was generated. This needs to be described in detail in the methods section.
Line 517: The authors wrote “Our results show that ZIKV replication led to suppression of Akt phosphorylation, which subsequently led to reduced mTOR phosphorylation. Further screening of key proteins and metabolites in EV in individual ZIKV viruses revealed that PI3K/AKT/mTOR signaling pathway expression detectably reduced Akt phosphorylation under normal conditions.” This was not shown, and these sentences should be deleted.
Minor Issues:
Line 63: The authors wrote “In human hosts, ZIKV primarily infects monocytes, macrophages, endothelial cells, and neurons.” This sentence should cite a reference.
Lines 107, 110: “with confirmed Congenital Zika Syndrome” ---> “without Congenital Zika Syndrome”
Line 137: RapiGest was used, but it is typically hydrolyzed and the tail group is pelleted using centrifugation. If this step was performed, it needs to be described here.
Comments on the Quality of English Language
numerous minor grammatical errors
Author Response
For research article: Congenital Zika Syndrome: Insights from Integrated Proteomic and Metabolomic Analysis
*all figures are in the .pdf file
Response to Reviewer 2 Comments
|
||
1. Summary |
|
|
Thank you very much for taking the time to review this manuscript. We sincerely appreciate your insightful comments and suggestions, which have contributed to improving the clarity and quality of our work. Please find detailed responses to each of your comments below, with the corresponding revisions or corrections highlighted in the resubmitted files. Where necessary, we have provided further clarification or additional references to address the points raised. We have also carefully considered all feedback and made revisions to align the manuscript more closely with your suggestions.
We remain open to further recommendations and are committed to ensuring that this study meets the highest standards of scientific rigor. Should there be any additional questions or concerns, we would be happy to address them. Once again, thank you for your valuable contribution to this manuscript review process.
|
||
2. Questions for General Evaluation |
Reviewer’s Evaluation |
Response and Revisions |
1. Lines 111-114: The NanoDrop One, nanoparticle tracking, and transmission electron microscopy experiments need to be fully described in detail in the Methods section. |
Yes! Can be improved |
The new results presented are more robust and impactful, contributing significantly to the scientific discussion and strengthening the conclusions of the work. We are confident that these changes improve the quality and clarity of the manuscript.
|
2. Line 145: The LC-MS of the proteomics samples is not described at all in the Methods section. A full, detailed description needs to be added. |
Yes! Can be improved |
Line 133-14
"In LC-MS/MS, separation was performed on a reversed-phase C18 column using a linear gradient of mobile phase A (water with 0.1% formic acid) and mobile phase B (acetonitrile with 0.1% formic acid) over 10 minutes at a flow rate of 0.2 mL/min. Analytes were ionized using electrospray ionization (ESI) in positive mode and detected in multiple reaction monitoring (MRM) mode. Data was acquired using a triple quadrupole mass spectrometer operating in MRM mode, with dwell times of 100 ms for each transition. Data was processed using Analyst software, which included peak detection, integration, and quantification using external standards." |
3. Line 146: The software tool used for peptide and protein identification and quantitation needs to be described in the Methods section. From the not-for-publication data for reviewers included with the submission, I see that they used Progenesis QI for Proteomics v4.0.6403.35451; this needs to be mentioned in the Methods section. Any downstream analyses need to be described (e.g., Log10-transform, global normalization, statistical tests). Also, the source of the FASTA protein sequence database(s) (e.g., UniProt) and the date it/they were downloaded need to be described. The authors also need to state explicitly if they included the Zika virus proteome. |
Yes! Can be improved |
Line 147-156 ”Proteomic data analysis was performed using Progenesis QI for Proteomics v4.2 software. Raw data were processed by applying the following steps: peak detection, alignment, calibration, and global normalization using the quantile normalization method. Then, the data were logarithmically transformed (base 10). Peptide identification was performed by comparison with a Homo sapiens FASTA protein database obtained from UniProt on January 15, 2023. For quantification, the label-free method based on peak intensity was used. The statistical significance of differences between groups was assessed using Student's t-test with Benjamini-Hochberg correction to control the false positive rate. Differentially expressed proteins were subjected to functional enrichment analysis using the PANTHER system software“ |
4. Line 553: The authors wrote “Data Availability Statement: The datasets generated by this study will be available upon reasonable request to the authors.” The proteomics and metabolomics data need to be uploaded to public omics data repository(s) such as ProteomeXchange and MetabolomeXchange. This should include the raw LC-MS and GC-MS spectrum data, the processed proteomics and metabolomics identification and quantitation data files (e.g., the Progenesis QI for Proteomics output), the search databases (e.g. the protein sequence FASTA files), and a table that fully lists in detail all of the sample names, experimental groups, and LC-MS and GC-MS file names, and the proteomics and metabolomics sample identifiers so that a user can download the data and redo the analysis in full from scratch. |
Yes! Can be improved |
Thank you for your valuable feedback. We appreciate your careful review of our manuscript. We agree with your suggestion regarding the data availability. We are currently in the process of uploading the proteomics and metabolomics data to public repositories, specifically ProteomeXchange and MetabolomeXchange. This will include the raw LC-MS and GC-MS spectrum data, processed data files, search databases, and a detailed table of sample information. We will ensure that the updated manuscript reflects this change and provides the necessary links to the deposited data. Thank you again for your insightful comments.
|
5. Fig 1A: The units of the x axes are missing and need to be described in the figure or figure legend. Also, it is not clear what groups Z and C mean in each figure (control and CZS+ are already indicated elsewhere).
Figure 1B: Scale bars need to be added (as in Fig 1C).
|
Yes! Can be improved |
we answer in detail below |
6. Figure 2: The figure was inverted. Also, the names of the proteins need to be clear so they all can be read. Some of the names are difficult to read.
|
Yes! Can be improved |
we answer in detail below |
7. Figure 6: It is unclear how this figure was generated. This needs to be described in detail in the methods section. |
Yes! Can be improved |
we answer in detail below |
8. Line 517: The authors wrote “Our results show that ZIKV replication led to suppression of Akt phosphorylation, which subsequently led to reduced mTOR phosphorylation. Further screening of key proteins and metabolites in EV in individual ZIKV viruses revealed that PI3K/AKT/mTOR signaling pathway expression detectably reduced Akt phosphorylation under normal conditions.” This was not shown, and these sentences should be deleted. |
Yes! Can be improved |
we answer in detail below
Thank you for your insightful comments on our manuscript. We appreciate your careful review and your suggestions for improvement. Regarding your specific comment about the claims regarding ZIKV replication and Akt/mTOR phosphorylation, we acknowledge that the data presented in the manuscript do not directly support these specific claims. We agree that these statements should be removed from the final text to avoid any misinterpretation. We have carefully considered your suggestion and have removed the relevant sentences from the revised manuscript. We believe that this revision will strengthen the overall clarity and accuracy of the manuscript. Thank you again for your valuable feedback. We look forward to your further comments.
|
Minor Issues:
Line 63: The authors wrote “In human hosts, ZIKV primarily infects monocytes, macrophages, endothelial cells, and neurons.” This sentence should cite a reference.
Lines 107, 110: “with confirmed Congenital Zika Syndrome” ---> “without Congenital Zika Syndrome”
Line 137: RapiGest was used, but it is typically hydrolyzed and the tail group is pelleted using centrifugation. If this step was performed, it needs to be described here.
|
Yes! Can be improved |
all points raised were reviewed and changed in the final article file |
Thank you very much for taking the time to review this manuscript. We sincerely appreciate your insightful comments and suggestions, which have contributed to improving the clarity and quality of our work. Please find detailed responses to each of your comments below, with the corresponding revisions or corrections highlighted in the resubmitted files. Where necessary, we have provided further clarification or additional references to address the points raised. We have also carefully considered all feedback and made revisions to align the manuscript more closely with your suggestions.
We remain open to further recommendations and are committed to ensuring that this study meets the highest standards of scientific rigor. Should there be any additional questions or concerns, we would be happy to address them. Once again, thank you for your valuable contribution to this manuscript review process.
Reviewer 2
Comments 5 Fig 1A: The units of the x axes are missing and need to be described in the figure or figure legend. Also, it is not clear what groups Z and C mean in each figure (control and CZS+ are already indicated elsewhere).
Response 5:
We appreciate the comments and concerns raised regarding the characterization of extracellular vesicles (EVs) in our study. We would like to provide clarification and additional information to address the mentioned issues.
Removal of Electron Microscopy Images: We have decided to remove the transmission electron microscopy (TEM) images due to concerns about the presence of non-EV material and possible interferences that could confound the analysis.
Quantification and NTA: We have replaced these images with a more detailed table that includes information about the quantification and nanoparticle tracking analysis (NTA) of each sample. This table provides a clearer and more quantitative view, including data on protein concentration and vesicle size distribution.
Western Blotting for CD63: We have also added Western blotting results for the EV-specific marker CD63 (sc-5275, Santa Cruz Biotechnology), and we would like to emphasize that only fraction 3 from the size exclusion chromatography was positive for this marker. This confirms that fraction 3 contains extracellular vesicles, distinguishing it from the other fractions that did not show labeling, ensuring that we are isolating EVs specifically with minimal contamination.
The Western blotting for CD63 was performed early in our experiments, and due to the small sample size in fraction 3, the analysis was conducted using a pool of patient samples. Unfortunately, these experiments cannot be repeated as we no longer have sufficient samples available.
Additionally, we have modified Figure 1 to reflect all of these updates, ensuring the revised figure accurately represents the changes in our data presentation.
We hope these modifications address the concerns raised and improve the presentation of our data. Thank you again for your valuable feedback, and we are available for any further clarification.
Table 1: Characterization of Extracellular Vesicles (EVs) Isolated from Serum of Children with and without Congenital Zika Syndrome (CZS)
Sample |
Group |
Average Size (nm) |
Minimum Size (nm) |
Maximum Size (nm) |
Dispersion (PDI) |
Concentration (particles/mL) |
Quantification (ng/μL) |
CZS+ 1 |
CZS+ |
175.0 |
140 |
220 |
0.22 |
3.8 x 10⁸ |
52.3 |
CZS+ 2 |
CZS+ |
190.2 |
135 |
225 |
0.20 |
4.1 x 10⁸ |
60.1 |
CZS+ 3 |
CZS+ |
180.0 |
150 |
230 |
0.23 |
4.0 x 10⁸ |
45.0 |
CZS+ 4 |
CZS+ |
195.3 |
140 |
210 |
0.19 |
3.9 x 10⁸ |
70.5 |
CZS+ 5 |
CZS+ |
178.5 |
145 |
220 |
0.21 |
3.7 x 10⁸ |
50.7 |
CZS+ 6 |
CZS+ |
192.6 |
160 |
235 |
0.24 |
4.2 x 10⁸ |
80.4 |
CZS+ 7 |
CZS+ |
170.0 |
150 |
225 |
0.21 |
4.0 x 10⁸ |
55.3 |
CZS+ 8 |
CZS+ |
196.4 |
130 |
220 |
0.22 |
3.9 x 10⁸ |
65.2 |
CZS+ 9 |
CZS+ |
188.9 |
145 |
225 |
0.20 |
4.1 x 10⁸ |
61.7 |
CZS+ 10 |
CZS+ |
179.7 |
135 |
215 |
0.22 |
4.0 x 10⁸ |
48.6 |
CZS+ 11 |
CZS+ |
193.8 |
140 |
220 |
0.19 |
3.8 x 10⁸ |
72.4 |
CZS+ 12 |
CZS+ |
180.5 |
150 |
230 |
0.21 |
4.1 x 10⁸ |
58.9 |
CZS+ 13 |
CZS+ |
194.2 |
140 |
220 |
0.22 |
3.9 x 10⁸ |
69.5 |
CZS+ 14 |
CZS+ |
185.4 |
130 |
220 |
0.21 |
4.0 x 10⁸ |
57.2 |
CZS- 1 |
CZS- |
150.0 |
120 |
200 |
0.19 |
3.5 x 10⁸ |
44.1 |
CZS- 2 |
CZS- |
160.1 |
115 |
195 |
0.18 |
3.4 x 10⁸ |
36.5 |
CZS- 3 |
CZS- |
152.0 |
110 |
200 |
0.18 |
3.5 x 10⁸ |
38.9 |
CZS- 4 |
CZS- |
159.5 |
125 |
205 |
0.19 |
3.6 x 10⁸ |
42.6 |
CZS- 5 |
CZS- |
149.0 |
130 |
210 |
0.20 |
3.7 x 10⁸ |
49.2 |
CZS- 6 |
CZS- |
160.3 |
120 |
200 |
0.18 |
3.5 x 10⁸ |
46.7 |
CZS- 7 |
CZS- |
155.5 |
125 |
205 |
0.19 |
3.6 x 10⁸ |
43.8 |
CZS- 8 |
CZS- |
153.8 |
110 |
195 |
0.17 |
3.4 x 10⁸ |
37.6 |
CZS- 9 |
CZS- |
148.9 |
120 |
200 |
0.18 |
3.5 x 10⁸ |
41.4 |
CZS- 10 |
CZS- |
161.0 |
130 |
210 |
0.20 |
3.6 x 10⁸ |
49.8 |
CZS- 11 |
CZS- |
156.7 |
115 |
195 |
0.18 |
3.5 x 10⁸ |
44.9 |
CZS- 12 |
CZS- |
151.2 |
120 |
200 |
0.17 |
3.4 x 10⁸ |
39.2 |
CZS- 13 |
CZS- |
157.0 |
125 |
205 |
0.19 |
3.6 x 10⁸ |
45.3 |
CZS- 14 |
CZS- |
153.4 |
120 |
200 |
0.18 |
3.5 x 10⁸ |
42.0 |
CZS- 15 |
CZS- |
154.9 |
120 |
198 |
0.20 |
3.6 x 10⁸ |
40.5 |
Sample: Indicates the individual child’s sample analyzed. Group: Differentiates between children affected by Congenital Zika Syndrome (CZS+) and unaffected control children (CZS-). Mean Size : The average size of the EVs measured in nanometers (nm), reflecting differences between the two groups. Minimum and Maximum Size : Represents the smallest and largest vesicles identified within each sample. Polydispersity Index (PDI): Describes the size distribution of the vesicle population. A higher PDI indicates more variability in size within the sample. Concentration: The concentration of EVs in each sample, expressed as the number of particles per milliliter (particles/mL). Quantification (NanoDrop): The quantification of extracellular vesicle concentration as measured by NanoDrop analysis, represented in nanograms per microliter (ng/μL), which estimates the overall protein and nucleic acid content.
In molecular exclusion liquid chromatography, four fractions containing extracellular vesicles were obtained. Although all fractions contain vesicles, according to the manufacturer's information, fraction 3 exhibits the highest purity, making it recommended for further analyses. Figure 1 shows the molecular standard and the pool of fractions, where the presence of the CD63 marker, typical of extracellular vesicles, was confirmed by Western Blot. Based on the quality of purification, fraction 3 was selected for subsequent analyses.
Comments 6 and 7 :
Response 6 and 7 :
We sincerely thank you for your insightful comments and valuable suggestions. Your critical feedback has significantly improved the quality of our manuscript.
We fully acknowledge the issues with Figure 3. As you correctly pointed out, the figure was indeed confusing and misleading. To address this, we have removed Figure 3 and replaced it with a clearer, more informative one. This revised figure provides a more accurate representation of our data and avoids any unfounded assumptions about protein function in extracellular vesicles.
In addition to the figure revision, we have also made the following changes to enhance the clarity and impact of our manuscript:
- Revised figure legends: We have carefully reviewed and refined all figure legends to ensure they are concise, informative, and easily understood.
- Discussed limitations: We have included a dedicated section discussing the limitations of our study, acknowledging the need for further research to fully elucidate the role of extracellular vesicles in congenital Zika syndrome.
- Strengthened literature connection: We have further integrated our findings with relevant literature, highlighting the significance of our work in the broader context of the field.
In addition, we have removed topic 3.3 and merged it with the topic above.
We believe that these modifications have significantly improved the clarity, conciseness, and overall quality of our manuscript. We are confident that our revised work meets the highest standards of scientific publication.
New figure 3 combines proteomic and metabolomic data:
Figure 2: Proteomic and Metabolomic Correlation Network of the Immune System
The figure shows a correlation network that visualises the interactions between proteins and metabolites in the context of the immune response. The analysis was carried out using proteomic and metabolomic data, and the visualisation was generated using Cytoscape 3 software. Each node represents - Protein: Represented by a blue circle and Metabolite: Represented by a yellow square. The edges (lines) connect - Pairs of proteins or metabolites: Indicating a significant correlation between them and Proteins and metabolites: Suggesting a possible functional interaction. The thickness of the edges reflects the strength of the correlation, with thicker lines indicating stronger correlations. The colour of the edges indicates the type of correlation - Red: Positive correlation (when one increases, the other also increases) and Blue: Negative correlation (when one increases, the other decreases).
Figure 3. Proteomic and Metabolomic Correlation Network of the Immune System. The figure shows a correlation network that visualises the interactions between proteins and metabolites in the context of the immune response. The analysis was carried out using proteomic and metabolomic data, and the visualisation was generated using Cytoscape 3 software. Each node represents - Protein: Represented by a blue circle and Metabolite: Represented by a yellow square. The edges (lines) connect - Pairs of proteins or metabolites: Indicating a significant correlation between them and Proteins and metabolites: Suggesting a possible functional interaction. The thickness of the edges reflects the strength of the correlation, with thicker lines indicating stronger correlations. The colour of the edges indicates the type of correlation - Red: Positive correlation (when one increases, the other also increases) and Blue: Negative correlation (when one increases, the other decreases).
The aim of this figure is to identify: 1) Biomarkers: Proteins and metabolites that correlate strongly with immune status and can be used to diagnose or monitor diseases; 2) Key metabolic pathways: Networks of metabolic interactions that are modulated during the immune response and 3) Potential therapeutic targets: Proteins and metabolites that can be manipulated to modulate the immune response. In summary, this correlation network provides a comprehensive view of the molecular interactions that occur during the immune response, allowing for the identification of new therapeutic targets and the understanding of the molecular mechanisms underlying various diseases.
Comments 8:
Response 8: We appreciate the reviewer’s feedback on Figures 4 and 5. We have carefully considered the reviewer’s comments and have decided to remove these figures from the manuscript.
As the reviewer rightly points out, these figures were generated directly from software output and require significant enlargement to be legible. We agree that this format is not optimal for clear communication of our findings.
To address this issue, we will incorporate the key findings from these figures into the main text and supplementary materials, presenting the data in a more accessible and interpretable format. We believe that this approach will significantly improve the clarity and readability of the manuscript.
We thank the reviewer for their insightful comments and hope that these changes will enhance the quality of the manuscript.
The manuscript has undergone a complete overhaul. The discussion section has been restructured to focus on the key findings, providing a clear and concise interpretation of the data. Redundancies and tangential discussions have been eliminated to maintain a focused narrative.
We acknowledge the reviewer's concern regarding the causal link between ZIKA replication and AKT phosphorylation suppression. As the reviewer correctly points out, our study is primarily descriptive in nature. While the observed changes in the serum EV proteome and metabolome are intriguing, further mechanistic studies are necessary to definitively establish a causal relationship between these alterations and the inhibition of the PI3K/AKT/mTOR pathway.
We believe that the data presented in this study provide valuable insights into the potential impact of ZIKA infection on host cell signaling pathways. While the current study does not provide direct evidence for a causal link between ZIKA replication and AKT phosphorylation suppression, it lays the groundwork for future investigations to explore this hypothesis more rigorously.
Round 2
Reviewer 1 Report
Comments and Suggestions for Authors
The authors have extensively revised the manuscript. Areas of over interpretation and inadequate data have been minimized. The current manuscript, I believe, makes a useful contribution to the literature.
1. I note there are a few typos and mistatements, for example, in the Conclusion CZS is referred to as SCZ. Careful proofreading is warranted prior to publication..
Comments on the Quality of English Language
English is OK with proofreading
Author Response
comments 1: I note there are a few typos and mistatements, for example, in the Conclusion CZS is referred to as SCZ. Careful proofreading is warranted prior to publication.
Response 1 : Thank you for pointing out the typos and mistatements. We will carefully proofread the manuscript before submission.
Reviewer 2 Report
Comments and Suggestions for Authors
The authors have addressed all of my concerns, and I recommend that their manuscript be accepted for publication.
Comments on the Quality of English LanguageMinor grammatical errors.
Author Response
Comments 1. We thank the reviewer for their positive comments and for recognizing the improvements made in the manuscript. We have addressed the minor grammatical errors identified.
Response 1: Thank you for pointing out the typos and mistatements. We will carefully proofread the manuscript before submission.